# Research citations building trust in Wikipedia: Results from a survey of published authors

**Carlos Areia**[1], **Kath Burton**[2]*, **Mike Taylor**[3], **Charles Watkinson**[4]

**1** Digital Science, University of Coventry, Coventry, United Kingdom, **2** Routledge, Taylor & Francis, London, United Kingdom, **3** Digital Science, University of Wolverhampton, Wolverhampton, United Kingdom, **4** University of Michigan Press, Ann Arbor, Michigan, United States of America

* katherine.burton@tandf.co.uk

## Abstract

The use of Wikipedia citations in scholarly research has been the topic of much inquiry over the past decade, however little is known regarding perceived Researchers trustworthiness of Wikipedia citations and representation of their work. This cross-publisher study (Taylor & Francis and University of Michigan Press) aimed to investigate author sentiment towards Wikipedia as a source of trusted information.

### Methods

A short survey was distributed to 40,402 authors of papers cited in Wikipedia (n=21,854 surveys sent, n=750 complete responses received). The survey gathered responses from published authors in relation to their views on Wikipedia's trustworthiness in relation to the citations to their published works. The unique findings of the survey were analysed using a mix of quantitative and qualitative methods using Python, Google BigQuery and Looker Studio.

### Results

Overall, authors expressed positive sentiment towards research citation in Wikipedia and researcher engagement practices (mean scores >7/10). Sub-analyses revealed significant differences in sentiment based on publication type (articles vs. books) and discipline (Humanities and Social Sciences vs. Science, Technology, and Medicine), but not access status (open vs. closed access).

### Conclusions

This study provides unique insights into author perceptions of Wikipedia's trustworthiness. Further research is needed to deepen the understanding of the benefits for researchers and publishers including academic citations in Wikipedia.

**Data availability statement:** The data underlying the results presented in the study are available from Figshare, https://doi.org/10.6084/m9.figshare.26037646.v2

**Funding:** The author(s) received no specific funding for this work.

**Competing interests:** Carlos Areia and Mike Taylor are employed by Digital Science (owner of Altmetric and Dimensions). Kath Burton is employed by the Taylor & Francis Group. Charles Watkinson is employed by the University of Michigan. Any ideas expressed here are the authors' own and do not represent the opinions of their employers.

# Introduction

Studies determining the extent to which Wikipedia citations appear in books and journal articles have questioned the appropriateness of using Wikipedia as a source [1] in academic research while others have noted the value of engaging with Wikipedia more generally in combating the spread of misinformation [2]. Other studies have analysed the use of Wikipedia citations in open access publications [3], how Wikipedia citations in Altmetric Explorer might determine an institution's reach [4] or how Altmetric Attention Scores might offer researchers an alternative to more typical metrics within specific fields of research (e.g., cancer) [5]. Our study was designed around collecting information from authors whose work has been cited in Wikipedia. By design, the study was limited (to publications in the Taylor & Francis (T&F) and University of Michigan Press (UMP) portfolios), one-way (academic citations in Wikipedia) and observational (soliciting opinions from published authors via survey). As such, the results we have reported are primarily concerned with opinions from T&F and UMP authors whose work has been cited in Wikipedia. By deliberately choosing to focus on soliciting author perceptions about trust in relation to citations to their work, our study adopted a blend of quantitative and thematic analysis to explore authors' responses and free-text comments about the perception of Wikipedia. Our study offers a unique insight from the perspective of authors at two research publishers which both complements and departs from existing studies into academic citations in Wikipedia.

The growing body of literature about "academic Wikipedia" [6] suggests that the relevance of Wikipedia citations in relation to research reach and potential impact is growing. Furthermore, publications exploring the accuracy of Wikipedia citations within scientific publications continue to increase (see Fig 1 below, source Altmetric); illustrating the important role of Wikipedia as an information literacy tool [7]. However, the extent to which citations in Wikipedia offer value to academic researchers is a lesser-known facet of the interest in academic Wikipedia, one that we explore in more detail through the analysis of survey responses.

## Context of the 'academic Wikipedia' community and combating spread of misinformation

As the Wikimedia Foundation also notes, there is a "thriving movement" [8] in community sentiment towards improving the accuracy and trustworthiness of articles in Wikipedia. The community sentiment towards improving the accuracy of Wikipedia is reflected by the number of academic communities running edit-a-thons and dedicated conferences that are now emerging to connect scholars, professionals and the wider Wikipedia community [9].

Movements within the library and information studies community that support engagement with, and interrogation of Wikipedia are also increasing. Writing in Leveraging Wikipedia: Connecting Communities of Knowledge [10] Merillee Proffitt highlights the role of

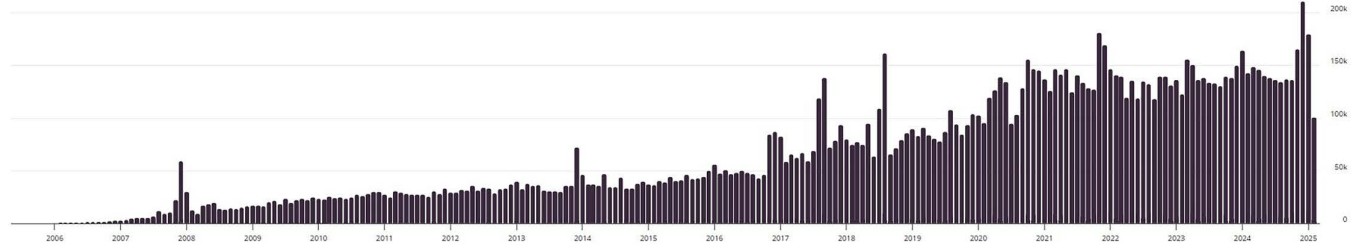

**Fig 1. Wikipedia Citations in Scientific Publications (Source: Altmetric accessed 20 June 2023, WikipOI publication years 2014-2023).**

the "Wikipedian" - volunteer editors and supporters of Wikipedia – as central to ensuring that information held in Wikipedia remains credible. A role that has extended to residencies within libraries [11] and at publishers [12], indicating the significance of editing not only to improve the accuracy of source citations in publicly available information, but as important research and learning tool.

While initiatives like Wikithons and Residencies are useful engagement activities, the EZProxy Access now provided to a select group of Wikipedia Editors via The Wikipedia Library (and adopted by a growing number of scholarly publishers) is increasing the number of research citations to validated publications in Wikipedia. While the provision of access to vetted Wikipedia Editors via The Wikipedia Library (TWL) is instrumental in increasing the addition of accurate research citations, the effect of Open Access (OA) publication on citation accuracy in Wikipedia is an increasingly significant factor in establishing trust. Yang et al, 2023 in Wikipedia and Open Access [13] found that OA publication is likely to increase citation in Wikipedia, specifically for articles that have been recently published and yet to accrue many citations.

A 2020 published study by Nicholson et al [14] posed the question in relation to the>6.6m articles now published in Wikipedia (as of June 2023): "just how reliable are the sources cited by Wikipedia articles, particularly with respect to scientific topics?" The study found that research citations in Wikipedia "are more than twice as likely to be supported than the scientific literature in general" and that "the citation context offers a more complete picture, potentially affecting decisions by everyday readers and choices of editors" [14]. The findings from these two studies offer a powerful endorsement of Wikipedia's role in increasing the accuracy of citation to validated research in a publicly available source of information. Yet, there are no studies assessing the published authors' thoughts on Wikipedia citations to their articles or books.

Wikipedia plays a major role as an audience interface. As such ensuring Wikipedia's accuracy is of great importance and citation verifiability is key to Wikipedia's credibility as a trusted source information, but guidelines for the use of Wikipedia citations are outdated. Wikipedia demands reliable, published sources, e.g., peer-reviewed books and journal articles. Through initiatives like The Wikipedia Library (TWL), academic publishers are opening up their content to trusted TWL editors to add verifiable citations to Wikipedia records.

During 2022/23, a group of data scientists and scholarly publishers drawn from Digital Science, Routledge, Taylor & Francis and University of Michigan Press designed and conducted an observational study into researchers' views on the accuracy of Wikipedia citations to their published works. This study set out to gather author sentiment towards Wikipedia as a trusted source of information. And explore how citation in Wikipedia might help combat misinformation in the context of increasing public engagement with and access to validated research. What value this presents to the researcher were key considerations of the study.

## Study objectives

To help explore this, the study set out the following primary and secondary objectives:

Primary: To understand researchers' views on the accuracy and trustworthiness of Wikipedia in representing the outcomes of their research to a broader audience.

Secondary:

1. Explore whether there are any significant differences between researchers' views on the accuracy and trustworthiness of Wikipedia in representing the outcomes of their research across different uses of Wikipedia types (e.g., research, teaching, etc.).

2. Explore whether there are any significant differences between researchers' views on the accuracy and trustworthiness of Wikipedia in representing the outcomes of their research across different disciplines.

3. Explore whether there are any significant differences between researchers' views on the accuracy and trustworthiness of Wikipedia in representing the outcomes of their research across different OA status of the cited publications.

4. Narratively explore and report free text survey answers.

5. Obtain a pool of researchers who we can follow up with to explore the topic of trust in Wikipedia and public engagement in more detail.

## Methods

### Design

This is a prospective observational study, with a cross-sectional data review of Wikipedia content and prospective contact of selected wiki pages cited researchers, referred to as "participants" from this point forward in the manuscript.

The survey was constructed in both Alchemer Survey Platform, Louisville, United States (T&F dataset) and Qualtrix XM, Washington, United States (UMP) and a sample set of responses produced in each system prior to launch in order to set up a script to run the code for future analysis. The survey participants were contacted directly with an invitation to complete a survey via link embedded in the email. There was no open platform for anyone to participate in the survey and there was no mandatory requirement to complete the survey. No incentives were offered. Survey participants were informed that the survey was for research purposes only and processed for analysis by the authors of the survey. Authors were informed that anonymised responses may be shared externally, but no personal data would be shared. Links to both T&F and UMP privacy policies were also shared to inform participants of their privacy entitlements. The survey was launched on 9 December 2022 and closed on 24 January 2023. The study followed the CROSS reporting guidelines (see S1 Appendix. CROSS Checklist for further details).

### Data collection

Using Dimensions (Digital Science, London, UK) and Altmetric (Digital Science, London, UK) data, an initial dataset was built to identify Wikipedia mentions from a pre-defined list of publications. This list was created using the following rules:

- Published by Taylor and Francis Group (T&F) or University of Michigan Press (UMP)

- Published in the last 10 years

One Wikipedia mention per publication prioritised: by the most recent English language Wikipedia mention. If there were no English language Wikipedia mentions then the most recent citation was used.

According to Altmetric data a total of 3,966,439 published outputs are cited by Wikipedia. The study prioritised citations in English Wikipedia and targeted 40,402 of the published items cited by Wikipedia articles. The survey therefore targeted about 1% of the Wikipedia cited content available in Altmetric.

This dataset contained the following categories

**Publication:**

1. Publication ID

2. Publication title

3. Publication year

**Wikipedia related:**

4. Wiki language

5. Wiki URL

**Researchers:**

6. Researcher ID

7. First and last name

8. ORCID

9. First publication year

**Disciplines:**

10. FoR-based 6 disciplines (Social Sciences, Humanities, Medical and Health Sciences, Life Sciences, Physics and Mathematical Sciences, Engineering and Technology). Note that papers can have more than one discipline.

## Participant contact

Using the dataset, the Altmetric team provided both T&F and UMP with respective DOIs of the cited publications. Both T&F and UMP proceeded to extract the respective list of authors for each publication and to invite them to respond to our short survey. For each publication, we picked a single citing Wikipedia page (prioritised by the most recent English language Wikipedia mention, and if there were no English Wikipedia mentions, the most recent one) for the authors to examine and comment on.

Each survey participant was informed that the data collected during this optional (non-mandatory) survey would be anonymised and used for the purposes of analysis. Participants were invited to share their contact details if they wished to be contacted after the survey had completed. No personal data was extracted from the survey participants' responses and all survey responses were stored in a secure part of the project leads' servers only accessible to the project leads.

## Survey

The survey consisted of seven questions, one multiple choice, four being a Likert scale (scoring from 'strongly disagree', 0, to 'strongly agree', 10), one binary (Yes/No), and one question in free-text (Table 1). The data was collected between December 2022 and January 2023. The questions were distributed over no more than two pages systems used to circulate the survey.

## Free-text analysis

The survey design incorporated an optional free-text question (Q7). Placed at the end of the survey, this question was designed to solicit additional comments from respondents about the choices provided in the previous questions in the survey.

**Table 1. Survey questions.**

| N | Question | Type |
|---|----------|------|
| 1 | For what purposes do you use Wikipedia in your own research? Choose all options that apply. | Multiple choice |
| 2 | The citation accurately represents the content of my publication. | Likert (0–10) |
| 3 | This Wikipedia page is a reasonable representation of its subject. | Likert (0–10) |
| 4 | I would recommend this Wikipedia page to a colleague in the field. | Likert (0–10) |
| 5 | I would recommend this Wikipedia page to someone not in the field who wishes to understand this subject. | Likert (0–10) |
| 6 | Would you be willing to be contacted again and to be part of a bigger project on Wikipedia trust and public engagement? | Binary (Yes/No) |
| 7 | Anything to add about your response? | Free text |

In order to evaluate the responses, we developed a form of thematic analysis (after Braun and Clarke [15]). We interpreted the themes based on prevalence (i.e., an inductive interpretation of the comments provided) to focus our understanding on the meaning and significance of the free-text comments provided from the perspective of the participants.

Adapting Braun and Clarke's six phases we: read all the quotes (familiarising ourselves with what had been noted); generated some initial codes; sorted the codes into themes; grouped themes together to give the groupings some order; reviewed those groupings for coherence across those themes (did they make sense and not overlap too much); and finally reviewed for validity (are there enough quotes to interpret or are these just quotes). Following this model of identifying and interpreting patterns or themes within the free-text comments we developed a simple framework for evaluating the qualitative data derived from the survey based around the assignment of a high-level code and descriptive definition as follows:

High-level grouping of comments based on similarity (e.g., "The main reason I would not recommend this is because it is too short.")

Descriptive coding and definitions created based on high-level grouping (e.g., Inaccurate Information: wrong image, wrong data, incorrect citation, typos)

The 10 descriptive codes and their associated definitions that were applied to the qualitative data received are listed in Table 2.

The free-text Q7 was an optional part of the survey and did not require any data from respondents for them to complete the survey. Some respondents chose to add some simple comments (e.g., no, thank you). These comments were coded as "Irrelevant to our study"

**Table 2. Free-text coding and definitions applied to Question 7 descriptive responses.**

| Code | Definition [example] |
|------|---------------------|
| Irrelevant to our study | thank you, no, I'm too busy |
| Unable to comment fully | unfamiliar language, lack of subject expertise |
| Inaccurate information | wrong image, wrong data, incorrect citation, typos |
| Inaccurate emphasis | lack of objectivity, not well written, impact of study mis-represented, could do better, could include a different citation as research has moved on |
| How to use Wikipedia | I don't understand…; how do I edit; should I include citations |
| Editing the Wikipedia entry | specific ask to please correct citation, spelling, typos |
| Personal engagement with Wikipedia | I support them, I am an editor, I tell my students… |
| Benefits of Wikipedia | Summary, connect to public, |
| Disbenefits of Wikipedia | perpetuates inaccuracies, poor citation practice |
| Serious/legal implications | slander accusation, trigger warnings missing |

to simplify the qualitative analysis and accounted for around 2% of all free text responses provided.

## Data and statistical analysis

### Data analysis

The initial dataset to identify relevant manuscripts and Wikipedia pages was built using Altmetric and Dimensions data inside Google BigQuery (GBQ) and Google Cloud, Google, California, United States. Surveys were built and shared to participants using Alchemer (for T&F) and Qualtrics (for UMP). Final survey datasets were then aggregated using R [16] and the tydiverse library [17]. Only fully completed surveys were considered for analysis, no data imputation was used. There was no weighting of items or propensity scores have been used to adjust for non-representativeness of the sample. No sensitivity analysis was conducted.

Data was explored both in R, GBQ and Looker Studio, Google, California, United States. Some visualisations were also explored using VOS Viewer, Leiden University's Centre for Science and Technology Studies, Netherlands. For clarity, Table 3 provides an example row of the dataset publicly available in Figshare https://doi.org/10.6084/m9.figshare.26037646.v2

### Statistical analysis

Descriptive statistics will be presented in raw counts, proportions, means and respective standard deviations, where appropriate. Inferential statistical analyses were conducted to test hypotheses, t-tests were employed to compare the means between two groups, while Analysis of Variance (ANOVA) was used to compare means among three or more groups. All statistics were conducted using R software [16] and tidyverse library [17].

For all sub-analysis, each group had to have at least 30 responses to be considered, otherwise it was excluded from the sub-analysis.

Table 3. Free-text coding example expanded.

|  | Column name | Data included for this example survey response |
|---|---|---|
| 1 | coding_1 | Inaccurate emphasis |
| 2 | completion_language | English |
| 3 | finished | TRUE |
| 4 | pub_discipline | Life Sciences |
| 5 | pub_OA | Closed |
| 6 | pub_publisher | Taylor & Francis |
| 7 | pub_type | article |
| 8 | publisher_type | T&F |
| 9 | q_citation_accuracy | 8 |
| 10 | q_free_text | My publication is a critical assessment of the subject, as such it first presents the subject, which is the only part that it is used in the wikipedia page |
| 11 | q_recommendation_colleague | 5 |
| 12 | q_recommendation_public | 6 |
| 13 | q_subject_representation | 10 |
| 14 | responder_country | Canada |
| 15 | responder_type | other |
| 16 | time_finished | 12/22/09 14:44 |
| 17 | time_started | 12/22/09 14:41 |

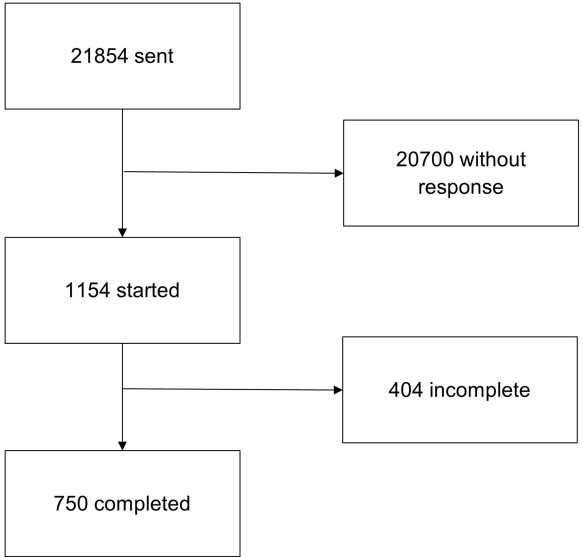

**Fig 2. Flowchart describing the steps in the survey distribution and response process.**

**Table 4. Breakdown of respondents by Fields of Research (FoR) aggregated discipline. To note is that one publication may be labelled as more than one discipline.**

| Discipline | Number of publications | Number of surveys received |
|---|---|---|
| Social sciences | 231 | 232 |
| Humanities | 221 | 225 |
| Life Sciences | 159 | 168 |
| Physical and Mathematical Sciences | 99 | 99 |
| Medical and Health Sciences | 87 | 90 |
| Engineering and Technology | 36 | 35 |

## Results

A total of 21,854 surveys were sent, 1154 started (5.3% response rate) and 750 completed (65.0% completion rate), this included 740 participants (Fig 2) from 60 different countries Fig 3), only completed surveys were included in the analysis.

A total of 727 unique publications had at least one author fully responding to the survey, a breakdown on the publication disciplines can be found in Table 4, and a more complex visualisation on the Fields of Research and keyword co-occurrence in Figs 4–6 shows a timeline of Wikipedia citations to the publications included in the survey.

Most of the answers to Question 1 ("For what purposes do you use Wikipedia in your own research?") use Wikipedia for teaching (344; 46%) and research (208; 28%), this was consistent across disciplines (Fig 7):

The results of the survey indicate that the participants generally agreed with the statements in Question 2," The citation accurately represents the content of my publication." (mean = 7.65, SD = 2.56), Question 3, "This Wikipedia page is a reasonable representation of its subject." (mean = 7.03, SD = 2.78), and Question 4, "I would recommend this Wikipedia page to a colleague in the field." (mean = 7.43, SD = 2.27). However, Question 5, "I would recommend this Wikipedia page to someone not in the field who wishes to understand this subject.", was broadly supported, it received a slightly reduced agreement level (mean = 6.35, SD = 2.91).

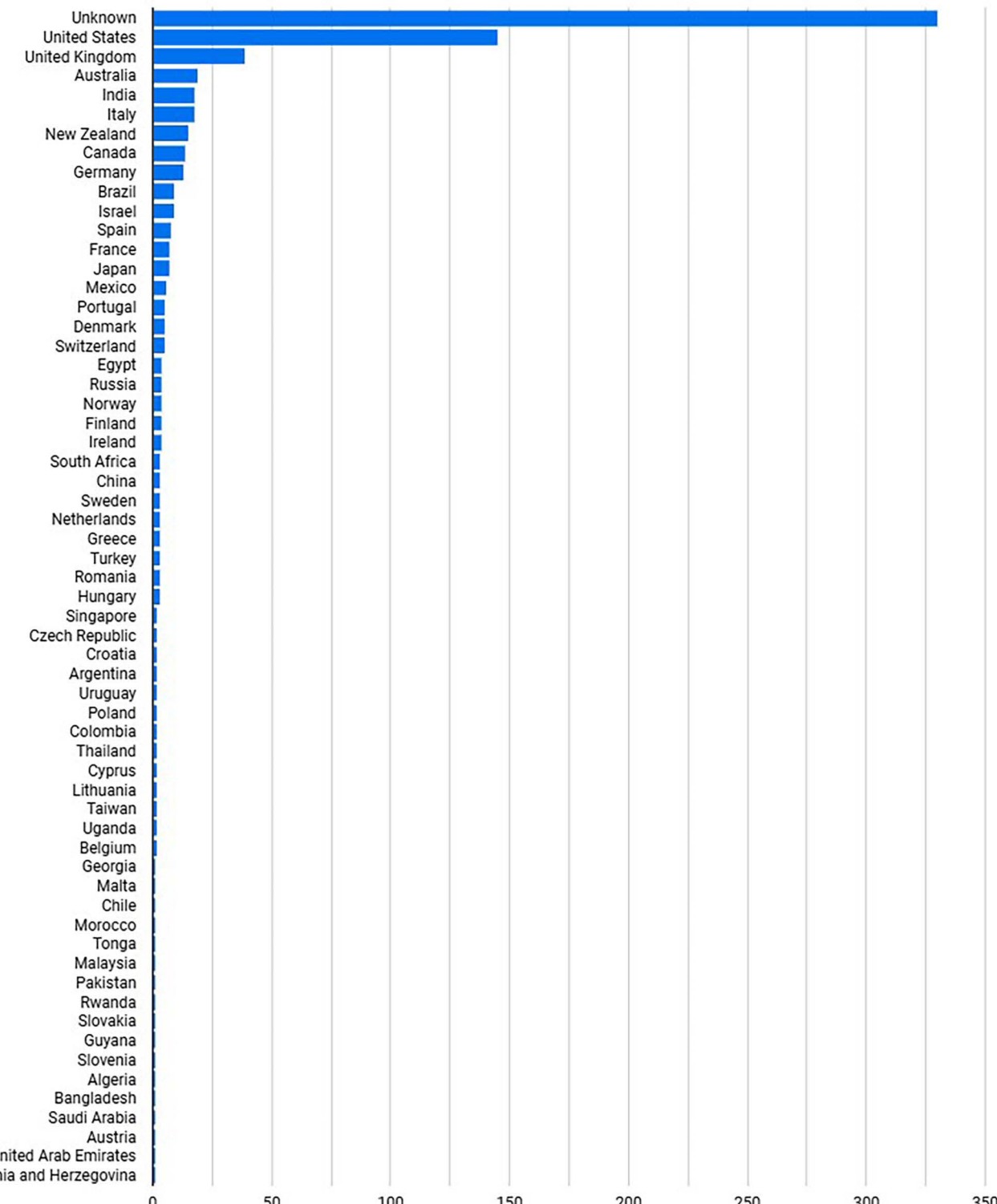

**Fig 3. Geographical Distribution of Respondents who successfully completed the survey.**

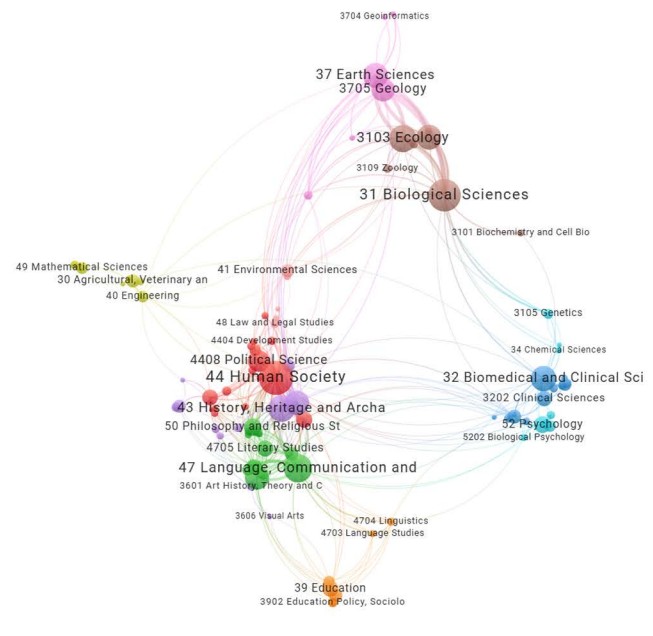
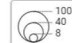

**Fig 4. Responses (pub_discipline) mapped to Dimensions Fields of Research (FoRs). Produced in VOS Viewer.**

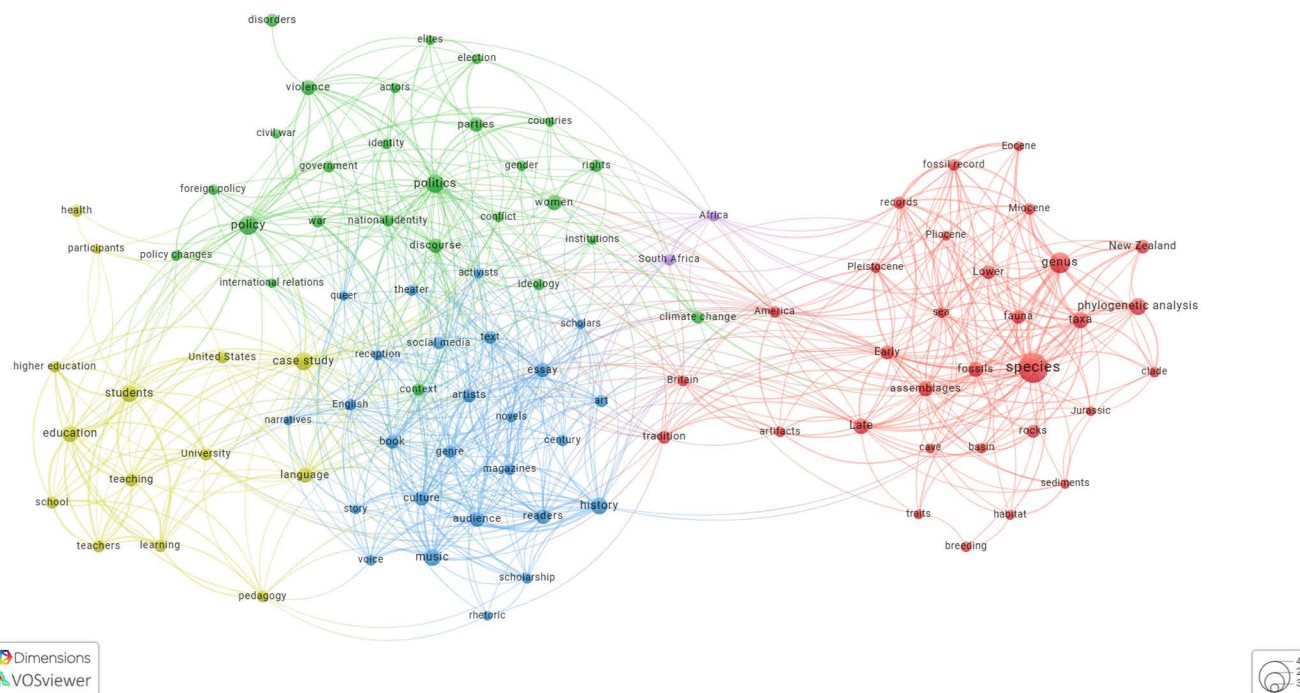

**Fig 5. Survey responses mapped to show keyword co-occurrences across the dataset.**

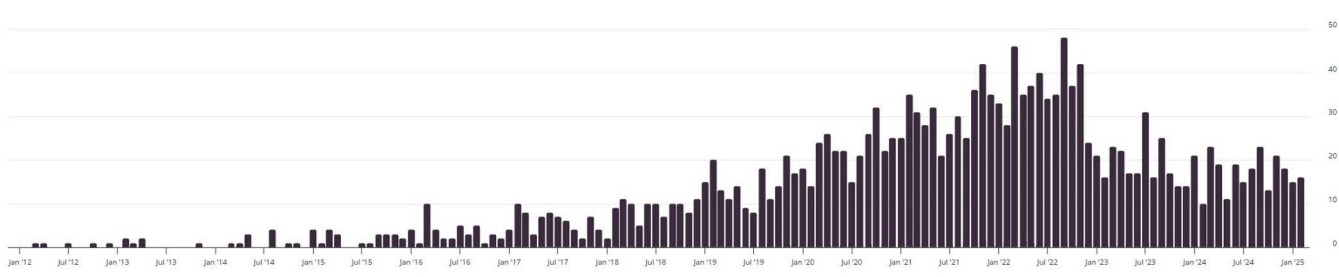

**Fig 6. A timeline of Wikipedia citations to the publications included in the survey.**

## Sub-analysis

### Responder use

Respondents using Wikipedia for Teaching consistently rated citation accuracy, subject representation, and recommendations more positively compared to those participating in Editathons (p = 0.0349 for citation accuracy, p = 0.0432 for subject representation, and p = 0.0213 for colleague recommendation). Teaching respondents were also significantly more likely to recommend Wikipedia to the public compared to Editathons (p = 0.0482). Research participants scored higher in public and colleague recommendations compared to Other groups (p = 0.0426 and p = 0.0311, respectively). In contrast, Editathons and None (blank responses) demonstrated lower confidence in Wikipedia's citation accuracy and broader utility.

### Papers versus books

These findings suggest that the type of publication may influence the recommendation scores, with the accuracy of citations to articles receiving higher public and colleague recommendation scores than those to books (p < 0.001). Subject representation was also statistically significant between articles and books, also favouring articles (p = 0.002). Citation accuracy was not different (Table 5).

### OA status

Our analysis does not show a statistically significant difference between Open and Closed publications. When analysing the different OA types, gold OA showed higher scores for most questions, however, none of these are statistically significant (Table 5). The OA status of publications referenced in the survey is included below for illustration purposes only.

### Disciplines

The dataset includes responses from six disciplines: Engineering and Technology, Humanities, Life Sciences, Medical and Health Sciences, Physical and Mathematical Sciences, and Social Sciences. Each discipline has a different number of responses, ranging from 35 for Engineering and Technology to 233 for Social Sciences (Table 5).

The mean public recommendation scores vary across disciplines, with Physical and Mathematical Sciences having the highest mean score (approximately 7.93) and Humanities having the lowest (approximately 6.7). Similar trends are observed for the mean scores of colleague recommendation, subject representation, and citation accuracy.

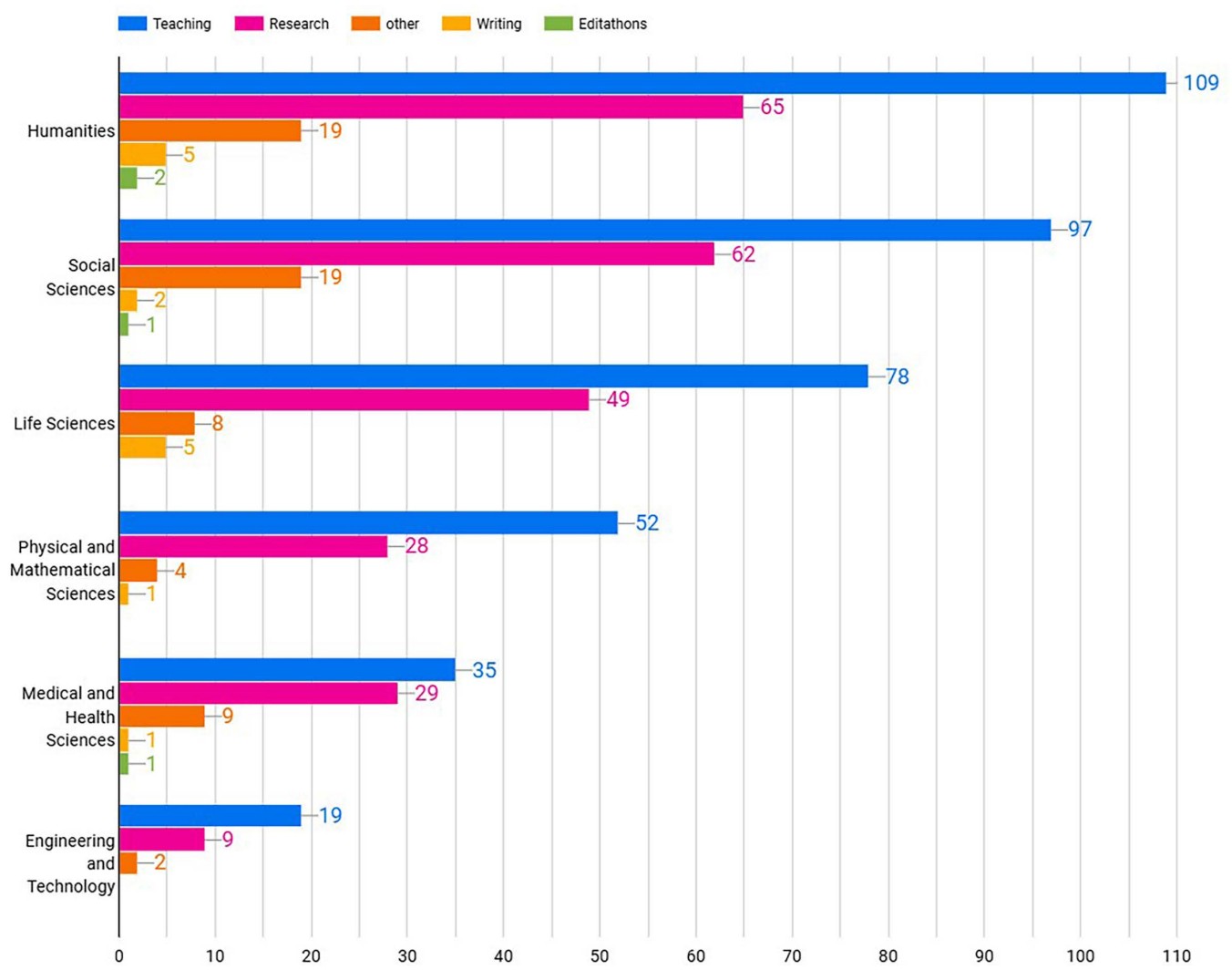

**Fig 7. Use of Wikipedia by Discipline.**

An ANOVA test was conducted to compare the mean public recommendation scores across the disciplines. The results indicate a statistically significant difference in the mean scores ($p < 0.05$). A Tukey HSD post-hoc test was then conducted to determine which specific disciplines have significantly different mean scores. The post-hoc test results highlight many significant differences. For example, it indicates that the Physical and Mathematical Sciences, when compared to Humanities and Social Sciences, have significantly higher public ($p < 0.05$) and colleague ($p < 0.001$) recommendation.

## Free-text results analysis

Approximately 24% of the participants' response were to the free-text question (Q7). Several comments received (5.2% of all responses) were irrelevant to our study (e.g., Thanks!) and have not been discussed any further. The largest proportion of viable categorised responses (4.4%) relate to the lack of source publication context ("inaccurate emphasis") in the

**Table 5.** Segmented results showing statistically significant difference in comparison between Responder Type, Article/Book, Open Access Status, and Disciplines a-l statistically significant between each other (p < 0.05), a-l* highly statistically significant between each other (p < 0.001).

| Group | N | Q2: Accuracy | Q3: Representation | Q4: colleague recommendation | Q5: public recommendation |
|---|---|---|---|---|---|
| **Responder Type** | | | | | |
| **Research** | 475 | 7.87±2.49 | 7.75±2.19 | 6.99±2.65 | 7.41±2.68 |
| Teaching | *345* | 7.96±2.46[a*,b] | 7.74±2.27[c,d,e] | 7.01±2.76[e,f,g] | 7.40±2.73[h,i] |
| Writing | 224 | 7.67±2.61[a*] | 7.58±2.34 | 7.00±2.79 | 7.33±2.74 |
| Other | 104 | 7.47±2.55[b] | 7.19±2.33 | 5.91±3.15[e] | 6.68±2.96 |
| Editathons | 35 | 8.08±2.13 | 7.80±1.89[c] | 7.09±2.61[f] | 7.51±2.63[h] |
| None | 116 | 7.16±2.97 | 6.66±2.62[d] | 4.46±3.12[g] | 5.84±3.17[i] |
| **Papers vs books** | | | | | |
| *Article* | 640 | 7.70±2.54 | 7.53±2.23[j] | 6.57±2.84[k*] | 7.24±2.68[l*] |
| *Book* | 100 | 7.34±2.69 | 6.88±2.48[j] | 5.32±3.10[k*] | 6.01±3.07[l*] |
| **OA Status** | | | | | |
| Closed | 474 | 7.59±2.57 | 7.45±2.35 | 6.47±2.94 | 7.15±2.82 |
| Open [all]: | 266 | 7.76±2.56 | 7.40±2.15 | 6.17±2.85 | 6.85±2.69 |
| Green | 112 | 7.84±2.66 | 7.27±2.36 | 6.15±2.91 | 6.79±2.84 |
| Hybrid | 70 | 7.87±2.35 | 7.33±2.11 | 5.88±3.13 | 6.64±2.92 |
| Gold | 55 | 7.58±2.51 | 7.77±1.81 | 6.77±2.47 | 7.31±2.14 |
| Bronze | 23 | 8.08±2.31 | 7.23±1.77 | 6.12±2.69 | 6.73±2.24 |
| Unlabelled | 6 | NA | NA | NA | NA |
| **Disciplines** | | | | | |
| SS | 232 | 7.74±2.37 | 7.38±2.13 | 5.85±2.94[m,n] | 6.70±2.73[r] |
| H | 225 | 7.39±2.77 | 7.13±2.37 | 5.95±2.90[o,p*,q] | 6.72±2.90[s] |
| LS | 168 | 7.86±2.89 | 7.68±2.24 | 6.84±2.89[m,o] | 7.39±2.80 |
| PMS | 99 | 7.91±2.28 | 7.86±1.99 | 7.32±2.49[n*,p*] | 7.91±2.23[r,s,t] |
| MHS | 90 | 7.48±2.90 | 7.44±2.72 | 6.91±2.93[q] | 7.32±2.84[t] |
| ET | 35 | 8.00±2.54 | 7.86±1.72 | 6.81±2.67 | 7.41±2.01 |

ET: Engineering and Technology, H: Humanities, LS: Life Sciences, MHS: Medical and Health Sciences, NA: Not Applicable, PMS: Physical and Mathematical Sciences, SS: Social Sciences

Wikipedia article. We discuss citation accuracy vs citation emphasis in the discussion section and include a summary analysis of the free-text responses in Fig 8.

## Discussion

The primary aim of this study was to collect and analyse researcher sentiment towards the Wikipedia citation of their article or book. Our results suggest there is general trust among researchers in Wikipedia both in terms of representativeness and accuracy. Most would also recommend the Wikipedia page where their work is cited to a colleague or the general public.

The survey questions were chosen to solicit as many responses as possible: ensuring that the survey was quick to complete was a major consideration. Expectations of the survey were that it would capture a range of views indicating how researchers engage with Wikipedia and situated the need for this study against the backdrop of a growing body of research about Wikipedia's uses and abuses, the growth in Wikipedia citations and the use of Wikipedia as a teaching and research tool especially in relation to increasing information literacy skills [18]. The response rate to our survey was 3.43%, with most responses engaging fully with both multiple choice and free-text survey questions within two weeks of the survey being launched.

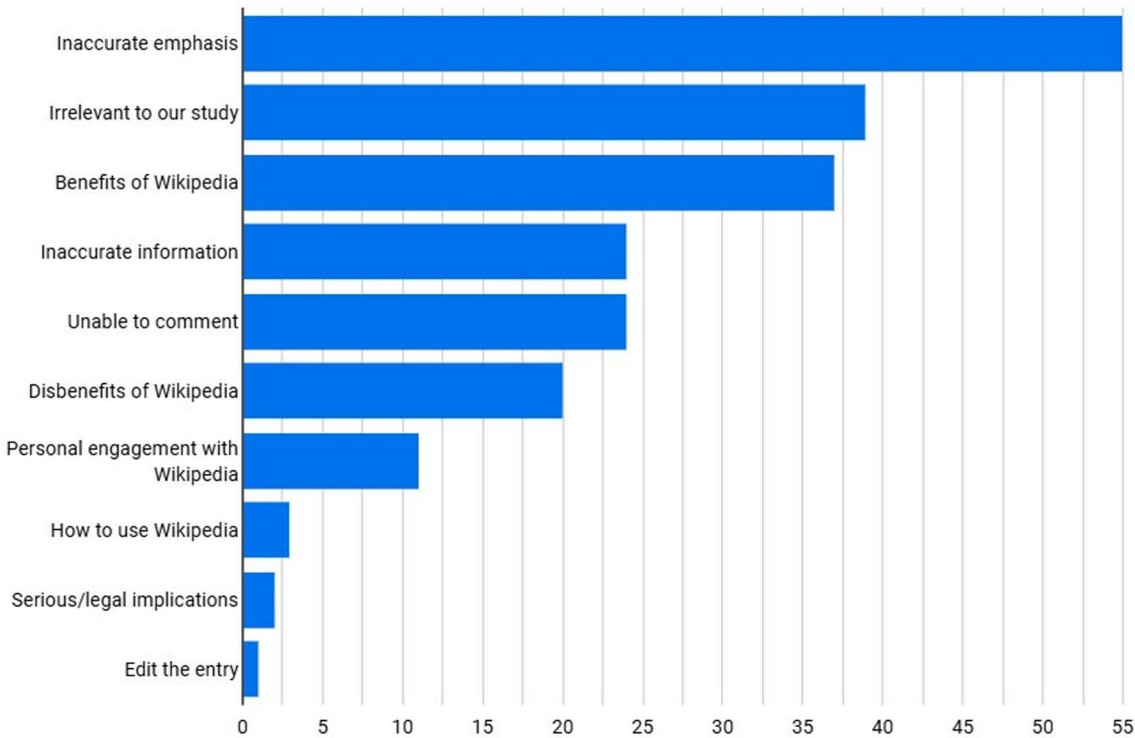

**Fig 8. Free-text results analysis.**

While the response rate alone was encouraging, the number of respondents who were willing to continue the conversation beyond the survey was notable. In response to Question 6: Would you be willing to be contacted again and to be part of a bigger project on Wikipedia trust and public engagement? Over 50% of respondents answered 'Yes' and provided contact details. As such the survey exceeded our expectations in gathering a secondary list of scholars with whom to explore trust in Wikipedia citations to their work. Although the survey design was designed to encourage completion, by being deliberately short (seven questions), the extent to which published authors would complete the survey was not anticipated.

As the main results indicate, researchers' views about the accuracy of the citations to their work, whether they would recommend the Wikipedia article about their work and if the article represented the original research topic well, were all positively indicated. There were interesting differences though between articles and books, with consistently higher scores for the article group throughout the survey (Table 4); with most being statistically significant.

In asking about the accuracy of the citation in Wikipedia, data from the survey does not reveal anything about the reliability or quality of the information in a book vs article. However, major studies indicate that researchers rate journal articles more highly as a source of trusted information [19].

A potential reason for lower scores in the book group is that book citations in particular are often to works that are only marginally about the subject of the Wikipedia article; even though they may include a term related to the article in the title. In other words, there is no evidence that the author of the Wikipedia article has read the whole book or if they have just skimmed selectively. This may be a product of reading behaviour, or it may reflect most of the citations being added by "professional" Wikipedia editors (i.e., who are being given access

via The Wikipedia Library) being generalists rather than specialists and therefore inserting citations based on fairly uncritical full-text searching. This lack of critical engagement with an entire book may also offer some indications for why the responses from humanities and social sciences authors attributed lower scores in their responses to public and colleague recommendation.

Exploring how much Wikipedia Editors engage in deep reading to construct the knowledge in the database or if they are primarily engaged in a "janitor" role [20], making sure some type of citation gets added, was beyond the scope of this study. However, the questions arising from the respondents' comments about "inaccurate emphasis" suggest that there is a high level of clean-up of Wikipedia records taking place, rather than a deep textual analysis. The guidance provided to Wikipedia editors [21] urging full engagement with source material before discussing that content in an article also suggests that the practices of Wikipedia editors are less engaged with the detail of source material than the citation to the source itself:

It's a small Wikipedia page and it's likely mostly correct but doesn't offer a lot of information. I'd have to dig deeper to know if it's all accurate or thorough. I do appreciate it citing one of the charts in my book, however! (Book Author)

Some respondents also noted that OA publication could be a contributing factor to increasing public engagement with published research. Overall, data from our survey did not substantiate claims that OA publication leads to greater citation via Wikipedia (as referenced in the in the Yang et al study [13] and because most articles cited were behind a paywall, see Table 4). The free-text results do reveal some researcher sentiment connecting OA publication to greater citation potential and that this would be a benefit to researchers, potentially enhancing their trust of Wikipedia:

It would be nice [if] Wikipedia cited articles were open to the public. (Journal author)

There are also some interesting differences when comparing discipline scores. Results suggest that among the six disciplines included in the publications covered by the survey, Physics & Mathematical Sciences (P&MS) are the most likely to refer a Wikipedia page to both a colleague and the public, more broadly, with Social Sciences and Humanities being the least likely to do the same. Together with Engineering and Technology, the P&MS group had also higher scores for perceived accuracy and representativeness of their research. The survey responses don't provide a clear indication as to why perceived accuracy in P&MS and Engineering and Technology is higher than other disciplines and there is little direct research evidence that indicates why perceived accuracy of Wikipedia citations would be higher in some disciplines over others. However, as research in P&MS and Engineering and Technology disciplines relies primarily on more quantitative and data-driven methodological approaches than research in the humanities, for example, the perceived accuracy may be attributable to researchers in those disciplines feeling more confident in the accuracy of a citation to their work that has been arrived at via more quantitative methods [22,23].

Analysing respondents' free-text entries in relation to the benefits perceived from Wikipedia citation, specifically where the respondent had scored 7 and above in relation to Q2 ("The citation accurately represents the content of my publication"), it becomes clearer that Wikipedia citation provides multiple benefits beyond citation accuracy. In the following sample quotes, the concept of trust is invoked in distinct ways:

Wikipedia seems to now be patrolled by trusted watchers, and it is possible to set alerts when content that may cite you or refer to you is altered - good way to check it yourself. (Journal author)

*Yes, I use Wikipedia in my research. I don't quote the articles themselves, but often use them to gain a broader understanding of a certain topic knowing that there are some caveats. The most important part of a Wikipedia page for me often is the references, which lead me to original source material. I'd rather use that than anything on Wikipedia. (Book author)*

As the option to add free text in response to Questions 2 and 7 was unrestricted by word count we also received a number of lengthy replies asking for edits to the Wikipedia entry and setting out in detail how the article should be re-written. The responses indicate that accuracy and purpose of citation in the Wikipedia entry are the main concerns of the respondents. Wikipedia citations to published works are topic driven and don't interrogate the topic in great depth but do add the validated source citation via DOI. The classification of the free-text comments indicates 77 "inaccurate emphasis" over 28 "inaccurate information"; respondents' free-text comments also back up the point that context is missing from the Wikipedia articles and that the purpose of Wikipedia is not as a primary source of information, but a "funnel" into that content.

In the context of research, Wikipedia is a funnel, not a source. That is, it helps finding in a single place an aggregate of references, with a commentary to facilitate their digestion. Ultimately, it's the peer-reviewed publications it cites that are the best guarantees of knowledge, and it's the responsibility of the researcher to adequately judge and evaluate each source they use. Consequently, it's important to understand the "trust" in Wikipedia as a gateway, more than as a final arbiter. (Journal Author)

Even when researchers/authors disagreed with the context (inaccurate emphasis) in which their work had been cited, adding research citations to Wikipedia, was thought to provide a beneficial entry point to validated research,:

*In research I will use it mainly to follow leads concerning individuals who come up elsewhere in my research - i.e., to see if they were prominent in some way. (Journal author)*

The benefits of the Wikipedia "funnel effect" for researchers were also not overlooked by researchers, with one respondent noting specifically how citations to their work in Wikipedia are developing relationships with the media to support more public engagement with their work:

*Thank you for alerting me to the link, citing my article. I am calling it to the attention of a journalist contact who is preparing to write on a related subject. (Journal author)*

Finally, some authors also commented that citation in Wikipedia could potentially extend access and increase visibility of their work to a broader public. While limited in the free-text responses, one respondent linked citation in Wikipedia to the potential for OA publication to increase public access to research:

*The citation shows the power of open access for topics of general interest. (Journal author)*

In summary, the free-text survey responses indicate a wide range of researcher sentiment about citation to their work in Wikipedia that can be broadly characterised as follows:

- Wikipedia seems to be a funnel into primary sources and rarely used to identify deep meaning about a topic (i.e., in comparison to a full, critical reading of the source material)

- Research citations added to Wikipedia may speed access to primary findings. This 'funnel effect' may be enhanced by the joint activities of researcher (i.e., through research and teaching activities) and TWL editor interventions (i.e., by adding and removing citations and writing articles based on validated research).

- Wikipedia research citations often distil global research, potentially overcoming regional biases and making access to and discussion of a broader range of global published research more equitable.

- Uses of Wikipedia in academia are increasing; the use of Wikipedia as a research and teaching tool supports the perception of Wikipedia as a trusted site of information

- Improving Wikipedia article and citation accuracy (i.e., researcher advocacy for Wikipedia) may reinforce the public message that they can trust the research being cited in Wikipedia

- Wikipedia citations may support public engagement with validated research and co-create researcher value in an increasingly open landscape

Due to the exploratory nature of our survey, we did not set out to gather evidence in support of a generalizable claim. However, our observations about the extent to which respondents engaged with the qualitative facet of the survey (and the insights captured) offer new models for perceiving how academic citations in Wikipedia are working towards building greater trust in the articles published and in Wikipedia more generally.

## Limitations

The research disciplines are categorised according to the manuscript Fields of Research classification and not as indicated by respondents. We acknowledge that, in some rare cases, there might be some respondents who are wrongly labelled in particular disciplines, however we strongly believe that our results rightly represent each respective field as most researchers publish within their own discipline.

The free-text responses, while a reasonably significant proportion of respondents and indicating the value of Wikipedia citations to researchers, have been analysed to determine the range of sentiments expressed by researchers. Further contextualisation of the benefits and disbenefits of Wikipedia citations requires more in-depth exploration via interviews and focus groups which were beyond the scope of the survey and subsequent analysis.. In addition, there is a chance that a biassed proportion of the respondents completed the free text responses to explore "negative" cases of the Wikipedia citation, such as "inaccurate emphasis", despite our quantitative results suggesting the majority agrees that the Wikipedia citations are accurately representing their research.

Finally, despite this being one of the largest cross-publisher studies conducted to date, our respondents were authors from only two publishers (T&F and Michigan), therefore we acknowledge our results may not be generalisable.

## Recommendations and future research

We acknowledge that this survey is only the first step in attempting to understand researchers' trust in Wikipedia citations to their work and would like to offer four recommendations for further consideration:

6) as the survey responses indicate, researchers perceived a high level of trust in the citation to their work in Wikipedia. Investing in further studies about how the inclusion of more accurate research citations in Wikipedia could enhance public trust in research

could further improve researchers' perception of the benefits of a citation to their work in Wikipedia.

7) as researchers are primarily engaging with Wikipedia as a teaching and research tool, there may be opportunities to encourage specialist researcher-authors to update Wikipedia articles where they have expertise and as such improve the accuracy of citations in Wikipedia. However, we recognise that the labour associated with monitoring the accuracy of and editing Wikipedia citations is not insignificant for already overburdened researchers.

8) as the study was conducted by a team of publishing and publishing-adjacent bibliometric organisations, exploring what shifts in the research and publishing landscape need to occur could further enhance the perception of Wikipedia as a trusted source of research information. Publishers and their bibliometrics, data-analytics partners can support a *shift in mindset.*

9) while publishers can make efforts to open their catalogues to vetted Wikipedia editors to increase accuracy of citations, some further investment in tools and system solutions that alleviate the burden on time-poor researchers is required.

Opening up catalogues to a broader range of published research is one measure that publishers can adopt to enhance researcher perceptions of trust in Wikipedia citations. More significantly, perhaps, is establishing the benefit of Wikipedia citations in the research evaluation process. Giving special consideration to where humans are involved in the creation, curation and distribution of knowledge and noting how "alignment" between humans and machines in the context of Large Language Models (LLMs) and the future of Wikipedia could continue to build trust [20]:

> *And if you appreciate the editors of Wikipedia are human, they have human motivations and concerns and that their motivations are providing high-quality educational material to align with your needs, then you can essentially put trust in the system.*

## Conclusion

The survey responses from researchers indicate that there is a reasonably high level of interest in the Wikipedia citations to their published work. Our study adds to the already existing literature by drawing on a unique dataset of published authors to determine sentiment in relation to Wikipedia citations to published works. Whereas previous studies have analysed Wikipedia citations to explore evidence of impact [24] our survey offered a unique approach to gathering author sentiment alongside quantitative analysis of academic citation in Wikipedia. The generally positive sentiment in both quantitative and qualitative responses indicates that being able to trust the accuracy of citations is of significant benefit to authors. Whereas findings from other studies, such as that conducted by Smith, McKinnell and Young [4] may play a role in extending institutional reach, the extent to which Wikipedia citations benefit individual authors (or their disciplinary field of research) is unclear.

In our study, however, where "inaccurate emphasis" had been introduced by the Wikipedia editors, researcher responses still indicated a level of trust in Wikipedia citations, primarily the citation serves as a funnel into the source material. This finding underscores the need for Wikipedia editors to continue adding research citations to articles and for initiatives like TWL to continue providing streamlined access to publishers' source content. While the survey results indicate that accurate citations added to Wikipedia contribute to its trustworthiness and enhance the value of Wikipedia as a funnel into research, a more intense interrogation of

the perceived researcher benefits of engaging with Wikipedia as a vital funnel into validated research is required.

Benefits that engage more deeply with Wikipedia as a vital funnel into validated research, especially where Wikipedia is to be perceived more widely as a trusted source of information for research and teaching.

Finally, the level of author engagement with both the survey and Wikipedia suggests that publisher collaboration with The Wikipedia Library may be beneficial to researchers, increase trust in Wikipedia more generally and provide additional benefits by opening up research to a broader public audience:

> *I think Wikipedia is a great site and am very glad to see it being taken seriously by publishers. It's a great space for public dialogue and presentation of ideas. (Book author, quote abbreviated)*

## Supporting information

**S1 Appendix. CROSS Checklist.**
(DOCX)

## Acknowledgements

The authors extend their gratitude to all survey participants for the time taken in responding to the questions. And to the survey and analytics teams at both Taylor & Francis and the University of Michigan Press.

## Author contributions

**Conceptualization:** Carlos Areia, Kath Burton, Mike Taylor, Charles Watkinson.

**Data curation:** Carlos Areia.

**Formal analysis:** Carlos Areia, Mike Taylor, Charles Watkinson.

**Investigation:** Kath Burton, Charles Watkinson.

**Methodology:** Carlos Areia, Kath Burton, Mike Taylor, Charles Watkinson.

**Visualization:** Carlos Areia.

**Writing – original draft:** Kath Burton, Charles Watkinson.

**Writing – review & editing:** Carlos Areia, Kath Burton, Mike Taylor, Charles Watkinson.

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
