## [Decision Letter · Decision Letter 0]

31 Oct 2024

PONE-D-24-41686Research Citations Building Trust in Wikipedia: Reults from a survey of published authorsPLOS ONE

Dear Dr. Burton,

Thank you for submitting your manuscript to PLOS ONE. After careful consideration, we feel that it has merit but does not fully meet PLOS ONE’s publication criteria as it currently stands. Therefore, we invite you to submit a revised version of the manuscript that addresses the points raised during the review process.

We look forward to receiving your revised manuscript.

Kind regards,

Jafar Kolahi

Academic Editor

PLOS ONE

Journal Requirements:

2. You indicated that ethical approval was not necessary for your study. We understand that the framework for ethical oversight requirements for studies of this type may differ depending on the setting and we would appreciate some further clarification regarding your research. Could you please provide further details on why your study is exempt from the need for approval and confirmation from your institutional review board or research ethics committee (e.g., in the form of a letter or email correspondence) that ethics review was not necessary for this study? Please include a copy of the correspondence as an ""Other"" file.

Additional Editor Comments (if provided):

Please consider the following list of improvements and suggestions to enhance the quality of the manuscript.

1. CROSS Checklist: Consider using the Consensus-Based Checklist for Reporting of Survey Studies (CROSS) to enhance the transparency and rigor of the survey study reporting.

2. Abstract Revision: Rewrite the abstract to clearly delineate the methods and results, adhering to a scientific style for clarity and conciseness.

3. Software/Tools Information: Ensure full details are presented for any software or tools used, including company names, cities, and countries, to provide transparency and allow reproducibility.

4. Sample Size Calculation: Include an explanation of the sample size calculation and any statistical methods used in the estimation to clarify the robustness of the sample.

5. Methods and Search Strategy: Describe the methods and search strategy used to compile the "pre-defined list of publications." This will provide clarity on the selection criteria and comprehensiveness.

6. Flow Diagram: Include a flow diagram outlining the steps taken to reach the total of 21,854 surveys, offering readers a clear visualization of participant selection.

7. Statistical Software: Specify the statistical software used for data analysis to ensure reproducibility.

8. Participant’s Characteristics Visualization: Use a map chart to display the geographic distribution of participants (740 participants from 60 countries). Additionally, use graphs to represent the academic affiliations and h-index of these participants.

9. Publication Analysis: For the analysis of “727 unique publications”, consider:

o Using VOSviewer (https://www.vosviewer.com/) to display hot topics via density visualization.

o The 727 unique publications cited by how many Wikipedia articles? analyze the topics of these Wikipedia articles. Utilize word clouds or other interactive graphics for a clear visualization of topics.

10. Limitations: Add a discussion of study limitations to provide a balanced perspective on the findings.

11. Future Research Topics: Based on the results, suggest some potential topics for future research, which can add depth and context to the study’s implications.

Reviewers' comments:

Reviewer's Responses to Questions

**Comments to the Author**

1. Is the manuscript technically sound, and do the data support the conclusions?

Reviewer #1: Yes

Reviewer #2: Yes

Reviewer #3: Partly

Reviewer #4: Yes

2. Has the statistical analysis been performed appropriately and rigorously? 

Reviewer #1: I Don't Know

Reviewer #2: Yes

Reviewer #3: Yes

Reviewer #4: Yes

3. Have the authors made all data underlying the findings in their manuscript fully available?

Reviewer #1: Yes

Reviewer #2: Yes

Reviewer #3: Yes

Reviewer #4: Yes

4. Is the manuscript presented in an intelligible fashion and written in standard English?

Reviewer #1: Yes

Reviewer #2: Yes

Reviewer #3: Yes

Reviewer #4: Yes

5. Review Comments to the Author

Reviewer #1: 1. It would be good to re-describe the research objectives in a more structured way. In particular, 2. 3 of secondary-objectives seems not to be academic research objectives. It would be all so desirable to add research questions.

2. In this study, presenting statistics by dividing free text analysis into 10 descriptive codes is not a good method. Through the Likert scale, the accuracy was shown as “mean = 246 7.65, SD = 2.56), but in the free text classification, “inaccurate emphasis” took up the largest proportion, so the two analysis results conflict. Therefore, the part that the paper wants to say becomes confusing.

3. It is unclear how the sub-group analysis was conducted. To understand Table 4, it is necessary to describe precisely how the research was conducted. After classifying whether the participants' papers are classified as papers/books, what type of OA they belong to, and what field they are in, were the statistics compiled based on the results of their own evaluations? You need to explain this specifically to the reader.

4. Since the participants are authors of the paper, most of them are researchers who are also providing education, it is inappropriate to distinguish them as “teachers and researchers” as in line 257

5. It is also necessary to present the reasons why social sciences and humanities are relatively undervalued.

Reviewer #2: Dear Authors,

After reviewing the manuscript, I would like to share some feedback. The study explores author sentiment towards Wikipedia citations in scholarly research, based on a survey of 750 authors across 60 countries. The findings indicate positive attitudes overall, but with significant differences between perceptions of books vs. articles and across disciplines. While the manuscript covers an important topic, there are areas that could benefit from further refinement. I encourage the authors to consider the following suggestions.

1. The abstract adequately introduces the study's purpose, methods, and findings, providing useful information overall. However, including specific percentages or statistics regarding positive outcomes could strengthen the abstract.

2. Although the topics discussed in the introduction are relevant to Wikipedia, the connection between the different sections is somewhat weak, indicating a need for more coherence in the structure. The text should be organized in a way that aligns with the main objective of the article and avoids deviating from the core topic.

3. The study benefits from a clear focus on ethical considerations and the use of validated data sources. However, the method of participant selection is not specified. Please clarify the sample size determination. Additionally, as the study only examines items published by two publishers (T&F and UMP), this limitation should be clearly acknowledged, as it may affect the generalizability of the findings.

4. Discussion is well-structured and covers important topics. However, the qualitative analysis relies heavily on the authors' observations, but it could be further strengthened by incorporating references from credible academic sources.

5. In conclusion, some sentences are lengthy and complex, which may hinder readability; simplifying them into shorter, clearer statements would enhance overall clarity. Addressing this concern would significantly improve the quality and impact of the conclusion.

6. Please consider utilizing the following references in your text where appropriate, as they can enhance your arguments and provide greater depth to your analysis:

 10.1002/asi.23694

 10.22034/ijism.2024.2008175.1196

 10.1108/OIR-02-2023-0084

 10.1080/0194262X.2016.1206052

 10.5195/jmla.2024.1730

 10.29024/joa.6

In some instances, the text appears to need additional citations, and integrating such references can strengthen the credibility of your findings and arguments.

Reviewer #3: The paper is well elaborated but some aspects should be considered by the authors and if possible, improved. For example, the results of the study have limitations due to low response rate. This raises questions about the generalizability of the findings. Also the study only targeted publications from Taylor & Francis and the University of Michigan Press, which could have affected the diversity of perspectives, especially in relation to fields of research that are less represented by these publishers.

The Survey Design was conducted in two different systems which, although aggregated later, may lead to inconsistencies in data collection.

While free-text responses were collected, the analysis was somewhat not so relevant. The study acknowledges that the free-text responses were only analyzed at a "surface level." Given the importance of qualitative data, this could be seen as a missed opportunity for deeper insights.

Disparities Across Disciplines: The study reveals significant differences in how Wikipedia citations are perceived across disciplines, particularly between the Physical and Mathematical Sciences (P&MS) and the Humanities., but the paper didn`t offer a strong explanation for why such disparities exist.

Although the study mentions that OA publication did not show statistically significant differences in trust compared to closed access publications, considering the relevance of open access in contemporary academic discourse, this aspect of the analysis feels underdeveloped and would have benefited from further exploration.

The study is based on survey responses collected over a short period. This not provide insights into how trust in Wikipedia citations may evolve over time.

The study relies entirely on survey data without incorporating other methods, such as in-depth interviews, focus groups, or case studies

While the paper concludes that there is general trust among researchers towards Wikipedia citations, this statement seems too optimistic given the significant discrepancies noted in the of trustworthiness between books and scientific papers, and across disciplines.

Reviewer #4: The submitted manuscript is of undeniable scientific interest. At the same time, the authors have followed all the conventions of manuscript structuring and established logical connections between paragraphs (all rhetorical moves have been taken into account, and all the signal and marker words used are functional and do not increase the wordiness of the text). The language of the manuscript is clear and well-justified. The authors have presented the manuscript as if they were perceiving it through the eyes of the reader. In other words, they anticipated all possible difficulties in understanding the information and explained them in detail in advance. This is one of those rare cases where it is difficult to suggest any recommendations for improving the manuscript. The only point that I believe requires a bit more clarification is the justification of the study's objectives.

This prospective observational study can serve as a good example for other authors in terms of structuring and transparency of information presentation. The described study was also planned and conducted in strict accordance with the design typical for studies of this kind. Data analysis enabled the authors to validate all the conclusions drawn. This study serves as an initiating step in the investigation of a complex and controversial process, which reflects the transforming paradigm of modern scientific communication. The framework for perceiving reality is shifting, and such studies are necessary for the international scientific community to be able to perceive scientific knowledge through a unified, consolidated understanding.

6. PLOS authors have the option to publish the peer review history of their article (what does this mean? ). If published, this will include your full peer review and any attached files.

**Do you want your identity to be public for this peer review?** For information about this choice, including consent withdrawal, please see our Privacy Policy .

Reviewer #1: No

Reviewer #2: No

Reviewer #3: No

Reviewer #4: **Yes: ** Elena Tikhonova

---

## [Author Response · Author response to Decision Letter 0]

10 Feb 2025

General Response to Editor and Reviewers

Thank you for your support and time dedicated to reviewing our manuscript. We thought your comments and suggestions greatly improved the overall quality of our analysis; we hope we have addressed all comments and suggestions satisfactorily.

Please find all responses below. 

Journal requirements

Response: Many thanks for double checking we are fulfilling all PLOS ONE’s style requirements. We have used PACE to check images and none were unacceptable. We can provide higher resolution images if required at typesetting stage.

2. You indicated that ethical approval was not necessary for your study. We understand that the framework for ethical oversight requirements for studies of this type may differ depending on the setting and we would appreciate some further clarification regarding your research. Could you please provide further details on why your study is exempt from the need for approval and confirmation from your institutional review board or research ethics committee (e.g., in the form of a letter or email correspondence) that ethics review was not necessary for this study? Please include a copy of the correspondence as an ""Other"" file.

Response: Thank you for your response. 

Additional Editor Comments

1. CROSS Checklist: Consider using the Consensus-Based Checklist for Reporting of Survey Studies (CROSS) to enhance the transparency and rigor of the survey study reporting.

Response: Thank you for your suggestion. Previously we had completed the CHERRIES checklist, however we have now changed this to the CROSS checklist and added to the manuscript as an Appendix, as well as referred to it in the methods section of the manuscript

2. Abstract Revision: Rewrite the abstract to clearly delineate the methods and results, adhering to a scientific style for clarity and conciseness.

Response: Thank you, we have done this.

3. Software/Tools Information: Ensure full details are presented for any software or tools used, including company names, cities, and countries, to provide transparency and allow reproducibility.

Response: Many thanks for your comment, we agree that it is important to outline all available information regarding the tools and software used. We have added more information on:

- Alchemy Survey Platform

- Qualtrics

- Dimensions and Altmetric (both part of Digital Science)

- Google Cloud, Looker Studio and Bigquery

- R (information already in the citation)

- Tidyverse library (information already in the citation)

- VOS Viewer

4. Sample Size Calculation: Include an explanation of the sample size calculation and any statistical methods used in the estimation to clarify the robustness of the sample.

Response: Thank you for your comment. We did not conduct a formal sample size calculation as we invited all eligible authors to participate in the survey. Another reason is that we tried to gather as many responses as possible to allow for sub-analysis.

5. Methods and Search Strategy: Describe the methods and search strategy used to compile the "pre-defined list of publications." This will provide clarity on the selection criteria and comprehensiveness.

Response: Thank you for your comment. The rules are already outlined below, the only requirements were:

- Published by Taylor and Francis (T&F) or University of Michigan Press (UMP)

- Published in the last 10 years

6. Flow Diagram: Include a flow diagram outlining the steps taken to reach the total of 21,854 surveys, offering readers a clear visualization of participant selection.

Response: Many thanks for the thoughtful suggestion, we agree this would be a great way to visualise the participant selection. This is now added in the beginning of the results section.

7. Statistical Software: Specify the statistical software used for data analysis to ensure reproducibility.

Response: Thank you for your suggestion. All statistics were conducted using R and the tidyverse package, now made clear in the relevant.

8. Participant’s Characteristics Visualization: Use a map chart to display the geographic distribution of participants (740 participants from 60 countries). Additionally, use graphs to represent the academic affiliations and h-index of these participants.

Response: Thank you for your suggestion. We agree that a country map would be visually appealing to readers. This has been added to the manuscript now showing the geographical distribution of the respondents. However, it will not be possible to represent the academic affiliations and h-index of these participants, the reason for this is that information was not collected during the survey. We will also not link these participants to any database to discover this information due to the anonymous nature of this analysis and other ethical considerations.

9. Publication Analysis: For the analysis of “727 unique publications”, consider:

o Using VOSviewer (https://www.vosviewer.com/) to display hot topics via density visualization.

Response: Thank you for your suggestion. Yes we have added Fields of Research and co-occurrence visualisations using VOS Viewer to the manuscript. That will hopefully shed some light to readers and provide context of the responses samples and publications being cited in Wikipedia part of this study.

o The 727 unique publications cited by how many Wikipedia articles? analyze the topics of these Wikipedia articles. Utilize word clouds or other interactive graphics for a clear visualization of topics.

Response: Many thanks for your suggestion. We agree with you that it would be interesting to further explore the Wikipedia citations of the included publications, so, we have added a timeline of Wikipedia citations for the included publications to the results.

10. Limitations: Add a discussion of study limitations to provide a balanced perspective on the findings.

Response: Thank you for your suggestion. You may now find the discussion of limitations in the respective “Limitations” sub-section of the Discussion.

11. Future Research Topics: Based on the results, suggest some potential topics for future research, which can add depth and context to the study’s implications.

Response: Thank you for this great suggestion, we have suggested potential topics for future research such as understanding how the inclusion of more accurate research citations in Wikipedia could enhance public trust in research

Reviewers' comments:

Reviewer #1

1. It would be good to re-describe the research objectives in a more structured way. In particular, 2. 3 of secondary-objectives seems not to be academic research objectives. It would be all so desirable to add research questions.

Response: Thank you for taking the time to review our manuscript and for your comments. We have now restructured the secondary objectives to academic research questions:

1-Explore whether there are any significant differences between researchers’ views on the accuracy and trustworthiness of Wikipedia in representing the outcomes of their research across different uses of Wikipedia types (e.g. research, teaching, etc…)

2-Explore whether there are any significant differences between researchers’ views on the accuracy and trustworthiness of Wikipedia in representing the outcomes of their research across different disciplines

3-Explore whether there are any significant differences between researchers’ views on the accuracy and trustworthiness of Wikipedia in representing the outcomes of their research across different OA status of the cited publications

4- Narratively explore and report free text survey answers.

5- Obtain a pool of researchers who we can follow up with to explore the topic of trust in Wikipedia and public engagement in more detail.

2. In this study, presenting statistics by dividing free text analysis into 10 descriptive codes is not a good method. Through the Likert scale, the accuracy was shown as “mean = 246 7.65, SD = 2.56), but in the free text classification, “inaccurate emphasis” took up the largest proportion, so the two analysis results conflict. Therefore, the part that the paper wants to say becomes confusing.

Response: Many thanks for your comment. We agree that it can be a bit confusing to the reader, however we do not agree we should not disclaim the free text results. In the text we do highlight the proportion of respondents that reported "Inaccurate emphasis” that accounts for a small proportion of all respondents. We have added a paragraph to the limitations to recognise this and that the free text responses can be biassed towards “negative” feedback.

3. It is unclear how the sub-group analysis was conducted. To understand Table 4, it is necessary to describe precisely how the research was conducted. After classifying whether the participants' papers are classified as papers/books, what type of OA they belong to, and what field they are in, were the statistics compiled based on the results of their own evaluations? You need to explain this specifically to the reader.

Response: Many thanks for your comment. Although we agree that the table contains a lot of information (as that is the main table, encapsulating most results inside the manuscript) we think we have made it clear in the methods section and objectives which comparisons were going to be conducted. Namely, Responder Type, Papers/Books, OA status, Disciplines. All comparisons were conducted within each of these, not comparing multiple classifications (we did not compare for example open access books versus closed access papers, for simplicity, and due to the included sample size, we have only compared open vs closed access and papers vs books). In addition, the dataset used is publicly available in Figshare (https://doi.org/10.6084/m9.figshare.26037646.v2), and statistical methods shown in the data and statistical analysis section, so readers can also explore the data and replicate the results if they choose to do so. However, and considering your suggestion to explain this to the reader in the manuscript, we have made the caption clearer and added further detail to the methods section showing an example response (part of the available data in Table 3). Hopefully this will give a clearer picture to readers how the sub analyses were conducted.

4. Since the participants are authors of the paper, most of them are researchers who are also providing education, it is inappropriate to distinguish them as “teachers and researchers” as in line 257

Response: Many thanks for your comment. You are correct in saying that the participants are authors of the paper, however “most of them are researchers who are also providing education” is an assumption. We have however noted an error in our code that instead of gathering the multiple selections from the datasets on the responder type it was only picking the main one (it was only capturing Teaching or Research even if the responder selected both). The sub-analysis has been completely re-done and the updated throughout the manuscript and available dataset.

5. It is also necessary to present the reasons why social sciences and humanities are relatively undervalued.

Response: Thank you for this comment. It is true that there were some differences in scores across disciplines. Responses from SSH disciplines showed lower scores in responses to public or colleague recommendation. However, we don’t consider this sufficient evidence of the social sciences and humanities being “relatively undervalued”. We have adjusted the text to reflect the specifics of the responses related to recommendations.

Reviewer #2

After reviewing the manuscript, I would like to share some feedback. The study explores author sentiment towards Wikipedia citations in scholarly research, based on a survey of 750 authors across 60 countries. The findings indicate positive attitudes overall, but with significant differences between perceptions of books vs. articles and across disciplines. While the manuscript covers an important topic, there are areas that could benefit from further refinement. I encourage the authors to consider the following suggestions.

Response: Many thanks for your time and consideration reviewing our manuscript. Your comments and suggestions greatly improved the quality of our results. Thank you again

1. The abstract adequately introduces the study's purpose, methods, and findings, providing useful information overall. However, including specific percentages or statistics regarding positive outcomes could strengthen the abstract.

Response: Thank you for this suggestion. We have added some statistics.

2. Although the topics discussed in the introduction are relevant to Wikipedia, the connection between the different sections is somewhat weak, indicating a need for more coherence in the structure. The text should be organized in a way that aligns with the main objective of the article and avoids deviating from the core topic.

Response: Thank you for this comment. We are not going to make significant changes to the structure, but have made some tweaks in response to this helpful comment.

3. The study benefits from a clear focus on ethical considerations and the use of validated data sources. However, the method of participant selection is not specified. Please clarify the sample size determination. Additionally, as the study only examines items published by two publishers (T&F and UMP), this limitation should be clearly acknowledged, as it may affect the generalizability of the findings.

Response: Thank you for your comment. We did not conduct a formal sample size calculation as we invited all eligible authors to participate in the survey. Another reason is that we tried to gather as many responses as possible to allow for sub-analysis.

Considering your suggestion we have added the following limitation:

“Finally, despite this being one of the biggest studies done to date, our respondents were authors from only two publishers (T&F and Michigan), therefore we acknowledge our results may not be generalisable.”

4. Discussion is well-structured and covers important topics. However, the qualitative analysis relies heavily on the authors' observations, but it could be further strengthened by incorporating references from credible academic sources.

Response: Thank you, we have adjusted where appropriate.

5. In conclusion, some sentences are lengthy and complex, which may hinder readability; simplifying them into shorter, clearer statements would enhance overall clarity. Addressing this concern would significantly improve the quality and impact of the conclusion.

Response: We have taken the above into account and adjusted where appropriate.

6. Please consider utilizing the following references in your text where appropriate, as they can enhance your arguments and provide greater depth to your analysis:

10.1002/asi.23694

10.22034/ijism.2024.2008175.1196

10.1108/OIR-02-2023-0084

10.1080/0194262X.2016.1206052

10.5195/jmla.2024.1730

10.29024/joa.6

In some instances, the text appears to need additional citations, and integrating such references can strengthen the credibility of your findings and arguments.

Response: We have reviewed the above suggestions and added references where appropriate.

Reviewer #3

The paper is well elaborated but some aspects should be considered by the authors and if possible, improved.

Response: We would like to thank you for your time reviewing our manuscript. Your comments and suggestions really contributed to the quality of our research.

For example, the results of the study have limitations due to low response rate. This raises questions about the generalizability of the findings. Also the study only targeted publications from Taylor & Francis and the University of Michigan Press, which could have affected the diversity of perspectives, especially in relation to fields of research that are l

---

## [Editor Report · Decision Letter 1]

17 Feb 2025

Research citations building trust in Wikipedia: results from a survey of published authors

PONE-D-24-41686R1

Dear Dr. Burton,

We’re pleased to inform you that your manuscript has been judged scientifically suitable for publication and will be formally accepted for publication once it meets all outstanding technical requirements.

Kind regards,

Jafar Kolahi

Academic Editor

PLOS ONE

---

## [Editor Report · Acceptance letter]

PONE-D-24-41686R1

PLOS ONE

Dear Dr. Burton,

I'm pleased to inform you that your manuscript has been deemed suitable for publication in PLOS ONE. Congratulations! Your manuscript is now being handed over to our production team.

Kind regards,

on behalf of

Dr. Jafar Kolahi

Academic Editor

PLOS ONE